# Metadata-Aligned 3D MRI Representations for Contrast and Sequence Understanding

## Abstract

Magnetic Resonance Imaging (MRI) offers diverse contrasts and acquisition protocols, yet the lack of standardized labels across sites and scanners makes automated sequence classification and contrast-aware applications challenging. We propose a metadata-guided CLIP framework for learning 3D MRI contrast representations by aligning images with their DICOM metadata. This alignment enables the model to capture both contrast-specific and acquisition-related variations, yielding embeddings that support diverse downstream tasks such as image–metadata retrieval and sequence classification, and can further serve as a foundation for contrast-invariant representation learning and cross-site harmonization. Evaluated on a large and heterogeneous clinical MRI dataset, our framework yields well-structured latent spaces, achieves strong image metadata retrieval, and forms meaningful unsupervised clusters of MRI sequences. Furthermore, the learned embeddings enable competitive few-shot sequence classification performance compared to fully supervised 3D networks. Code and weights are publicly available at [anonymised].

## 1   Introduction

Magnetic Resonance Imaging (MRI) is a versatile modality widely used in clinical practice, providing diverse contrasts and acquisition protocols that capture complementary anatomical and functional information. However, clinical MRI datasets are often highly heterogeneous, collected across multiple scanners, sites, and patient populations, and typically lack standardized sequence or contrast labels [1]. This variability poses major challenges for automated sequence classification[2], contrast-aware analysis[3], and downstream tasks such as image retrieval [2] or harmonization [4, 5]. Recent advances in self-supervised and contrastive representation learning, particularly CLIP-style frameworks, have shown that aligning different modalities can yield embeddings with strong generalization and transfer capabilities [6–9]. However, existing approaches often depend on full supervision or do not fully capture the rich acquisition metadata inherent to MRI due to limited data variability, leading to representations that remain sensitive to scanner and protocol specific variations [10–13].

We propose a metadata-guided CLIP framework for 3D MRI that aligns volumetric images with their DICOM [14] metadata to learn contrast-aware embeddings. These embeddings capture acquisition-specific variations, achieve strong image–metadata retrieval performance and form structured latent spaces that naturally cluster MRI sequences. Moreover, the learned representations enable competitive few-shot classification and provide a promising foundation for downstream tasks such as cross-site data harmonization and modality-aware image analysis.

Submitted to 39th Conference on Neural Information Processing Systems (NeurIPS 2025). Do not distribute.

## 2 Methods

MR-CLIP learns MRI contrast representations by contrastively aligning volumetric image embeddings with structured DICOM metadata (see Fig. 1). For each acquisition, a 3D image encoder extracts volumetric features, while a metadata encoder projects DICOM tags (via a natural language template) into a shared embedding space. To account for small parameter differences that do not meaningfully affect image contrast, we group metadata by binning numeric fields (e.g., TR, TE) and clustering categorical fields (e.g., Manufacturer), forming semantically similar acquisition groups and reduce 21,660 unique metadata combinations to 1,415 contrast labels, which are then used to guide the contrastive learning process. The list and distribution of used metadata are provided in the Appendix.

MR-CLIP is trained using a Supervised Contrastive (SupCon) Loss [15]. Let $z_i$ denote the anchor embedding for sample $i$, and let $P(i)$ be the set of positive embeddings for $i$, including exact matches and other samples from the same metadata group. The loss for anchor $i$ is

$$\mathcal{L}_i = -\frac{1}{|P(i)|} \sum_{p \in P(i)} \log \frac{\exp(z_i^\top z_p / \tau)}{\sum_{a \in A(i)} \exp(z_i^\top z_a / \tau)},$$

where $A(i)$ is the set of all embeddings in the batch excluding $i$, and $\tau$ is a temperature hyperparameter. This loss is calculated separately for image and metadata embeddings and then averaged. Compared to standard InfoNCE [16], which considers only a single positive per anchor, SupCon naturally handles multiple positives, encouraging the model to cluster semantically similar acquisitions. We also train a 2D variant of MR-CLIP that aligns individual slices with their corresponding metadata; in this case, the SupCon objective benefits from including different slices from the same brain, promoting consistent contrast representations and invariance to anatomical variations.

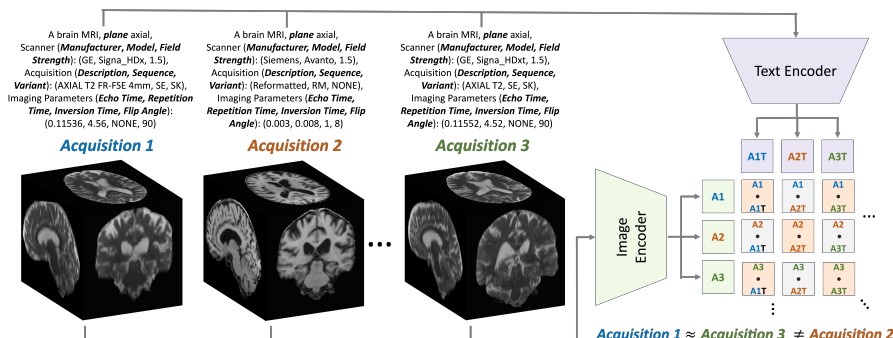

Figure 1: MR-CLIP aligns 3D MRI volumes with their corresponding DICOM metadata, resulting in contrast representations that are robust to anatomical variability and subtle parameter differences.

## 3 Results and Discussion

We evaluate MR-CLIP through three complementary experiments that assess cross-modal alignment, representation quality, and metadata interpretability. As summarized in Table 1, 2D and 3D MR-CLIP

Table 1: Cross-modal retrieval performance (%). Showing Recall@K (R@1/5/10) for image-to-text, 3D scan-to-text, and text-to-image retrieval. Linear classification accuracy (%) is shown in the rightmost column. Highest values in each column are bolded.

| Model | Image→Text | | | 3D Scan→Text | | | Text→Image | | | Linear Acc. |
|---|---|---|---|---|---|---|---|---|---|---|
| | R@1 | R@5 | R@10 | R@1 | R@5 | R@10 | R@1 | R@5 | R@10 | |
| BiomedCLIP | 1.4 | 5.0 | 8.4 | 2.5 | 9.8 | 15.0 | 3.6 | 9.5 | 13.1 | 39.0 |
| BiomedCLIP (Fine-tuned) | 50.0 | 78.5 | 82.6 | 67.4 | 89.1 | 92.1 | 38.5 | 65.8 | 71.8 | 75.5 |
| ViT-B/16 (InfoNCE Loss) | 65.6 | **85.2** | **90.4** | 68.8 | 92.2 | 94.4 | 49.3 | 69.3 | 76.6 | 71.3 |
| ViT-S/16 | 46.7 | 79.1 | 84.4 | 69.0 | 92.2 | 95.2 | 64.6 | 77.8 | 80.9 | 73.6 |
| **2D MR-CLIP** (ViT-B/16) | **66.0** | 77.3 | 78.3 | **78.7** | **94.2** | **95.3** | **90.9** | **93.6** | **94.4** | 82.6 |
| **3D MR-CLIP** | - | - | - | 60.2 | 79.0 | 82.0 | 79.3 | 91.6 | 94.0 | **86.9** |

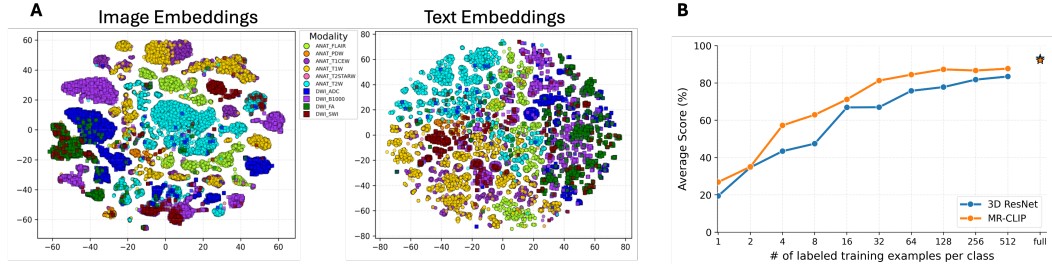

Figure 2: A: t-SNE visualizations of image and text embeddings, color coded by sequence. B: Few-shot learning performance of MR-CLIP, compared to supervied 3D ResNet baseline.

variants outperform baselines in image-to-metadata, metadata-to-image, and 3D scan-to-metadata retrieval (for 2D models, retrieval results are aggregated across slices to produce comparable 2.5D scan-level scores) across R@1, R@5, R@10, and in linear metadata classification. The 2D MR-CLIP achieves the highest overall retrieval scores, reflecting precise slice-level alignment and efficient feature utilization, while the 3D variant closely follows, demonstrating volumetric representations that generalize effectively across diverse imaging protocols.

To visualize the learned representation structure, we project image and metadata embeddings using t-SNE (Fig. 2A). MR-CLIP embeddings form distinct clusters across MRI sequence types, clearly separating anatomical and diffusion-weighted images. In few-shot sequence classification (Fig. 2B), MR-CLIP consistently outperforms a 3D ResNet baseline, particularly under low-shot settings (1–64 samples per class), while performing comparably when trained on the full dataset, highlighting its stronger inductive bias under limited supervision.

Finally, we analyze per-tag prediction accuracy under linear probing across 2D, 2.5D, and 3D MR-CLIP variants in Fig. 3. The 2.5D model performs best overall, suggesting that aggregating local slice context provides effective balance between efficiency and representational capacity. Discrete fields such as Acquisition Plane and Field Strength are classified with near-zero error, while numerical parameters (e.g., TE, TR) exhibit higher bin misclassifications but small average deviations, indicating predictions close to the true values.

Overall, MR-CLIP effectively disentangles image contrast from anatomical content, producing robust, contrast-aware embeddings that generalize across scanners and support downstream tasks such as retrieval, sequence recognition, and metadata analysis. Although grouping quality and metadata incompleteness and inconsistencies may introduce noise in training process, the framework establishes a scalable foundation for metadata-guided MRI representation learning, bridging image features with acquisition semantics for improved analysis and harmonization.

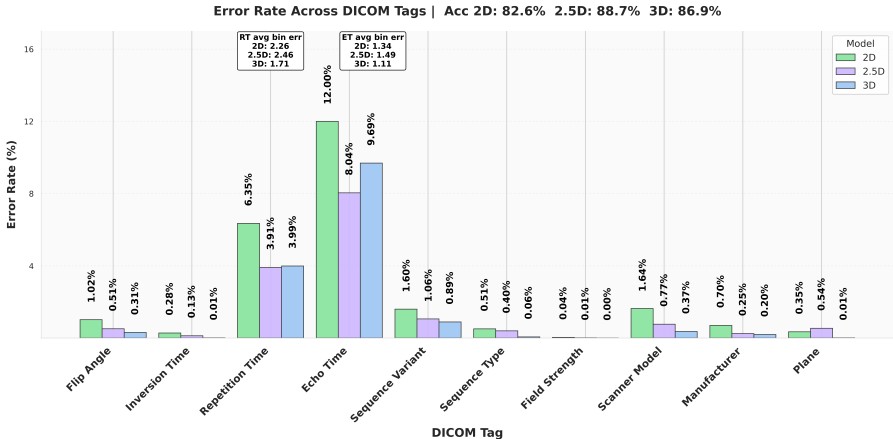

Figure 3: Error rates across DICOM tags based on linear probe classification results.

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

# A  Appendix

**Data**    Data usage approved under [anonymised]. Full list of used DICOM tags are given in Table 2 and distribution of tags are given in Fig. 4. All 3D MRI volumes are rigidly registered to the MNI template space and skull-stripped. From each registered volume, we extract a representative subset of slices by selecting every second slice from the central 100 slices, capturing the most diagnostically relevant anatomy while controlling dataset size. Acquisition plane (axial, coronal, sagittal) is determined from voxel resolution, with the highest-resolution dimension chosen as the slicing axis; for isotropic volumes, the axial plane is selected by default. Full pipeline is given in code repository.

**Implementation Details**    MR-CLIP is implemented in PyTorch and trained on three NVIDIA A100 GPUs (40 GB each) with a batch size of 3000 for 2D and 150 for 3D per GPU, using sharded loss as in the CLIP implementation [17]. Optimization uses Adam ($lr = 1e-4$, $\beta_1 = 0.9$, $\beta_2 = 0.98$) with weight decay 0.2, over 100 epochs with 2000 warm-up steps. Gradient checkpointing reduces memory usage, while patch dropout (0.5) and text dropout (0.2) are applied alongside standard image augmentations, including random affine transforms, resized crops, Gaussian blur, and horizontal flips. The codes are built upon OpenCLIP repository (`https://github.com/mlfoundations/open_clip`) (License provided in repository)

Table 2: DICOM metadata fields used in MR-CLIP for contrast and sequence representation learning.

| **DICOM Tag** |
| --- |
| Magnetic Field Strength |
| Manufacturer |
| Manufacturer's Model Name |
| Series Description |
| Scanning Sequence |
| Sequence Variant |
| Acquisition Plane (extracted from voxel size) |
| Echo Time (TE) |
| Repetition Time (TR) |
| Inversion Time (IR) |
| Flip Angle |

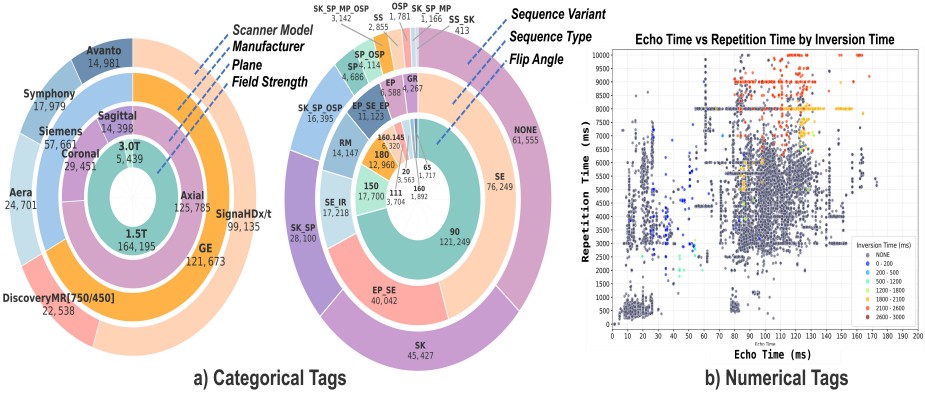

Figure 4: Overview of metadata distribution in our dataset. (a) Categorical tags including scanner, plane, field strength, and sequence information and flip angle. (b) Numerical distribution of echo and repetition times, color-coded by inversion time.

