# OpenReview forum: "Metadata-Aligned 3D MRI Representations for Contrast and Sequence Understanding"
_EurIPS.cc/2025/Workshop/MedEurIPS — EurIPS 2025 Workshop MedEurIPS Submission_

### Official Review · Reviewer_LEPc · 2025-10-24
**The authors present an interesting direction, but limited novelty and scalability**

**Rating:** 3
**Confidence:** 4

**Review:**

This paper presents a metadata-guided CLIP framework (MR-CLIP) that aligns 3D MRI scans with their associated DICOM metadata to learn contrast-aware representations. The motivation of the work is relevant in improving MRI representation learning through metadata alignment. The paper is generally well written and supported with reasonable experimental validation.

However, I find the overall methodological novelty to be limited. The approach largely builds on existing CLIP-style paradigms with minor adaptations for metadata fields and 3D MRI volumes. Several similar efforts have already explored modality alignment with acquisition parameters or structured text, and the contribution here feels incremental.

Overall, the paper is technically sound but falls short on conceptual novelty and practical scalability.

---

### Official Review · Reviewer_EEUs · 2025-10-29
**Interesting approach to scanner bias and multimodal embeddings**

**Rating:** 7
**Confidence:** 4

**Review:**

This paper addresses the dependency on scanner vendor, type and parameter on the appearance of MRI images, which is known to cause bias and generalization problems for AI models utilizing MR. To this end, the paper uses a CLIP-style model that aligns DICOM metadata with image embeddings, in an attempt to obtain contrast-aware embeddings.

This is a very interesting idea with high potential utility.

I am not sure I fully understand the approach taken:
- Would you not want your the representation of anatomy to be invariant to scanner parameters rather than aligned with it?
- Is it positive that different modalities are separated? (Maybe it is, but it's not clear to me)
- It is not clear to me what is predicted in Figure 3

While these points are unclear to me, I think they are mostly a consequence of page limit, not potential, and I would be very interested in discussing this paper at the conference. I therefore recommend acceptance (and perhaps a little clarification on predictive tasks and motivation in the final version as my questions suggest)

---

### Decision · Program_Chairs · 2025-10-31

**Decision:**

Accept (Poster)

**Comment:**

Both reviewers find the paper relevant and well written, addressing an important challenge of scanner bias and metadata utilization in MRI. One reviewer highlights its potential and discussion value, while the other points out limited methodological novelty and scalability. Given its conceptual relevance and likely value for discussion despite incremental innovation, the paper is accepted.